# Oxygraphy Versus Enzymology for the Biochemical Diagnosis of Primary Mitochondrial Disease

**DOI:** 10.3390/metabo9100220

**Published:** 2019-10-10

**Authors:** Matthew J Bird, Isabelle Adant, Petra Windmolders, Ingrid Vander Elst, Catarina Felgueira, Ruqaiah Altassan, Sarah C Gruenert, Bart Ghesquière, Peter Witters, David Cassiman, Pieter Vermeersch

**Affiliations:** 1Laboratory of Hepatology, Department of Chronic Diseases, Metabolism and Ageing, Katholieke Universiteit Leuven, 3000 Leuven, Belgium; isabelle.adant@kuleuven.be (I.A.); petra.windmolders@kuleuven.be (P.W.); ingrid.vanderelst@kuleuven.be (I.V.E.); catarina.amf@gmail.com (C.F.); david.cassiman@kuleuven.be (D.C.); 2Metabolomics Expertise Center, Center for Cancer Biology, CCB-VIB, 3000 Leuven, Belgium; bart.ghesquiere@kuleuven.vib.be; 3Clinical Department of Laboratory Medicine, University Hospitals Leuven, 3000 Leuven, Belgium; pieter.vermeersch@kuleuven.be; 4Department of Pediatrics, University Hospitals Leuven, 3000 Leuven, Belgium; 5Medical Genetics Department, King Faisal Specialist Hospital and Research Center, KSA MCD, Riyadh 43228, Saudi Arabia; altassanr@gmail.com; 6Department of General Pediatrics, Adolescent Medicine and Neonatology, Medical Center-University of Freiburg, Faculty of Medicine, 79106 Freiburg, Germany; sarah.gruenert@uniklinik-freiburg.de; 7Metabolomics Expertise Center, Department of Oncology, Katholieke Universiteit Leuven, 3000 Leuven, Belgium; 8Metabolic Center, University Hospitals Leuven, 3000, Leuven, Belgium; peter.witters@uzleuven.be; 9Department of Cardiovascular Sciences, Katholieke Universiteit Leuven, 3000 Leuven, Belgium

**Keywords:** Primary mitochondrial disease (PMD), enzymology, oxygraphy, oxidative phosphorylation (OXPHOS), respiration, diagnostics

## Abstract

Primary mitochondrial disease (PMD) is a large group of genetic disorders directly affecting mitochondrial function. Although next generation sequencing technologies have revolutionized the diagnosis of these disorders, biochemical tests remain essential and functional confirmation of the critical genetic diagnosis. While enzymological testing of the mitochondrial oxidative phosphorylation (OXPHOS) complexes remains the gold standard, oxygraphy could offer several advantages. To this end, we compared the diagnostic performance of both techniques in a cohort of 34 genetically defined PMD patient fibroblast cell lines. We observed that oxygraphy slightly outperformed enzymology for sensitivity (79 ± 17% versus 68 ± 15%, mean and 95% CI), and had a better discriminatory power, identifying 58 ± 17% versus 35 ± 17% as “very likely” for oxygraphy and enzymology, respectively. The techniques did, however, offer synergistic diagnostic prediction, as the sensitivity rose to 88 ± 11% when considered together. Similarly, the techniques offered varying defect specific information, such as the ability of enzymology to identify isolated OXPHOS deficiencies, while oxygraphy pinpointed PDHC mutations and captured POLG mutations that were otherwise missed by enzymology. In summary, oxygraphy provides useful information for the diagnosis of PMD, and should be considered in conjunction with enzymology for the diagnosis of PMD.

## 1. Introduction

Mitochondria are complex organelles serving a myriad of cellular functions including, but not limited to, calcium buffering, anaplerosis, cell death regulation, and ROS balancing. Their core function, however, is considered to be their energy generating capacity, and consequently they are often referred to as the “power houses” of the cell. At the core of this system is the oxidative phosphorylation (OXPHOS) system, a series of five complexes (CI–V) that collectively strip electrons from their donor (primarily the reducing equivalents NADH and FADH_2_) and store the energy in an electrochemical gradient of protons across the inner mitochondrial membrane, which is ultimately used by CV for the regeneration of ATP from ADP and inorganic phosphate.

Mitochondria are also unique in that their overall function, which depends on both the nuclear (nDNA, estimated at 1500 genes) and mitochondrial genomes (mtDNA, 37 genes) [1,2,3]. Further, mammalian cells contain hundreds of copies of their mtDNA genome per cell, which can give rise to a phenomenon known as heteroplasmy where a percentage of mtDNA only carry a pathogenic mutation, further complicating an already difficult diagnostic process [4].

Primary mitochondrial disease (PMD) can arise when any of more than the 250 known genes (and counting) have pathogenic variants [5], including in synergistically heterozygous patterns [6]. Ultimately, disruption and dysfunction may manifest in diverse mitochondrial aspects and functions, including the OXPHOS system, tricarboxylic acid cycle (TCA) cycle, and the mitochondrial structure. PMD can manifest at any age and the clinical presentation is diverse, but primarily affects the vital organs of the heart, skeletal muscle, liver, kidney, and brain, often with fatal or severely debilitating outcomes [7]. Further complicating their diagnosis are patients with secondary mitochondrial disease (SMD) [8]. SMD patients appear phenotypically and biochemically similar to PMD patients, but their underlying basis is non-mitochondrial. SMD patients have their roots in mutations in non-mitochondrial genes (e.g., *ATP7B* mutations causing Wilson’s disease) or environmental factors. While distinguishing PMD from SMD patients then is an ongoing challenge, this was not assessed in our own study.

Until the advent of next generation sequencing (NGS) technologies, molecular diagnosis of PMD was extremely challenging and primarily relied on enzymological testing of the OXPHOS complexes in patient fibroblasts or tissue biopsies (typically muscle) interpreted in conjunction with clinical indications of disease. While the use of NGS has transformed the diagnostic odyssey for many patients with PMD, biochemical diagnostics remain critical. For instance, in cases where NGS returns negative or indeterminate results (no variants in known PMD-associated genes identified, variants of unknown significance in known PMD-associated genes, or variants in genes for which no disease association is known), biochemical correlation or confirmation is essential. This complex diagnostic odyssey endured by patients was highlighted in one survey, which found that 48% of patients consulted more than six physicians before receiving a diagnosis, and 25% of patients saw more than 11 doctors [4].

In this regard, analysis of mitochondrial enzymes (primarily the OXPHOS system) by spectrophotometric methods (enzymology) has remained the gold standard biochemical test for detecting PMD. Surprisingly, although these methods have been well described for many years [9,10,11,12], the sensitivity and specificity of enzymological testing for the diagnosis of PMD is only sparsely reported in the literature as assessed in a panel of patients with defects of diverse genetic origins and pathophysiology. From the limited data available, the sensitivity in fibroblasts was in the order of 75% [13], and higher in muscle at 80–100% [13,14,15], albeit these numbers are drawn from limited sample sizes and are not paired in their analysis. Nonetheless, it is critical then to consider that fibroblasts as employed in our own study may not be the optimal sample for the diagnosis of PMD, and that PMD manifests with organ specific pathophysiology [16]. In this regard, it has been observed in a paired analysis that PMD is far more likely to be detected by the enzymology method in biopsies of affected organs such as skeletal muscle, heart, and liver than in fibroblasts [16]. Enzymology, it should also be considered, is time-consuming and labor-intensive unless highly sophisticated and automated systems are established, and optimal sensitivity requires an invasive tissue biopsy that should be analyzed fresh [17]. The number of individual tests required also grows as additional complexes are investigated, requiring five assays alone to capture the full gamut of possible primary OXPHOS dysfunction, with further tests required for defects such as in pyruvate dehydrogenase complex (PDHC), a commonly defective complex in cases of PMD.

We therefore questioned if the technique of oxygraphy applied in PMD fibroblasts might be a complementary tool for the diagnosis of PMD, offering a more rapid or sensitive biochemical test than enzymology. The technique itself simply measures the amount of oxygen being consumed in a sample in an airtight chamber and is thus a proxy for OXPHOS CIV activity. However, with the ability to subsequently inject multiple substrates and inhibitors into the assay, the activity of multiple systems can be indirectly inferred, including that of the TCA cycle and OXPHOS.

To this end, we sought to directly compare the sensitivity and disease specific information yielded through both enzymology and oxygraphy testing in a retrospective panel of 34 PMD genetically defined fibroblasts cell lines as compared against 12 control lines. Since both techniques provide a series of results (e.g., for different complexes), we then classified the probability of PMD based on the results as “unlikely”, “possible”, “likely”, or “very likely” using a scoring system.

## 2. Results

### 2.1. Patient Cohort

Genetically confirmed PMD patient and control fibroblast cell lines were retrospectively recruited from the UZ Leuven fibroblast bank (ethics approval number S60206), as described in Table 1. Our patient cohort is both clinically and genetically heterogeneous, to serve as a broad base from which to evaluate both enzymology and oxygraphy methods for the biochemical diagnosis of PMDs. The disease-causing genes (Figure 1) in our cohort were arrayed to a number of mitochondrial systems including primary OXPHOS mutations (CI, 7 patients; CIV, 2; CV, 1), mtDNA genes, and transcription translation and maintenance machinery (19), structural (1), and the TCA cycle (or closely associated, 4). Twenty-three of these mutations were encoded for by nDNA, and 11 by mtDNA (68% and 32%, respectively, of our cohort), and are therefore as reported, at least in adults, overly skewed towards nuclear defects [18].

It should be noted that this reported data was largely collected before the advent of NGS technologies which continues to uncover new nDNA encoded disease-causing genes at a great pace, and could be expected then to have slightly underestimated the prevalence of such mutations. The gender of the group was 20 females and 14 males. Where defined, the patients belonged to several disease groups including: Alpers (3), hypertrophic cardiomyopathy (1), Kearns-Sayre syndrome (1), Leigh syndrome (5), leukocencephalopathy with thalamus and brainstem involvement and high lactate (1), mitochondrial encephalomyopathy, lactic acidosis, and stroke-like episodes (MELAS) (4), Neuropathy, ataxia, and retinitis pigmentosa (NARP) (1), progressive external ophthalmoplegia (PEO)(2), and Sengers syndrome (1). The disease symptoms were similarly diverse, featuring many of the classic features of PMD, including: diabetes mellitus, exercise intolerance, lactic acidosis, myopathy, and neurological involvement such as ataxia or developmental delay (Table 1 and Appendix A). Similarly, the patients presented at diverse ages, with varying severities as indicated by their survival (ages 3–81, median 35 years) or age of death (0–72, median 17 years).

### 2.2. Enzymology

Enzymology in this article refers to the technique of measuring enzyme activities by spectrophotometric methods, where substrates or products linked to the respiratory chain complex of interest are monitored by absorbance over time, thus allowing the determination of the rate of activity. The technique is well established in clinical practice globally and considered the gold standard for the biochemical diagnosis of PMD. Testing is performed in fibroblasts, or more optimally in a primary tissue such as muscle (typically M. vastus lateralis) or liver [17]. Critically though, while the methods themselves are well described in the literature [10,11,12] and globally applied in a diagnostic context, there are only scarce reports of their sensitivity in populations of PMD patients [13]. To this end, we first sought to quantify this parameter in our cohort of genetically defined patients with PMD.

Primary or combined OXPHOS dysfunction most commonly impairs complexes I, III, and IV. For this reason, we focused our enzymological analysis on measuring the activity of the respiratory chain complexes CI–IV. CV and TCA cycle enzyme activities which are not routinely performed were omitted. Mutations directly affecting these systems only account for 17% of our patient population (including two patients with SLC25A42 mutations that are not classically monitored by enzymological methods; Table 1). Nonetheless, this could lead us to underestimate the sensitivity of the enzymological method, as described here.

Rates of CI–IV activity, and normalized rates to citrate synthase (CS; surrogate marker for mitochondrial volume) are displayed in Figure 2. The equivalent data, as well as a disease prediction (see Materials and Methods), are presented in Table 2.

Taken together, OXPHOS dysfunction was indicated in 68% of cases (“possible”, “likely”, or “very likely”). If CV and TCA cycle defects are excluded, considering our limited analysis of only respiratory chain complexes, the sensitivity rises modestly to 70%, indicating that their inclusion in our analysis does not broadly skew the results. Dysfunction also tracks relatively closely with the underlying defect, where the respiratory complex(es) predicted to be affected are most prominently impaired. Controls were indicated to be 100% “unlikely” by our scoring methods, and all control values followed as a normal distribution as tested by both the Shapiro–Wilk and Kolmogorov–Smirnov tests in Prism 8, where p ≥ 0.05 was considered to follow a normal distribution (Table 2). Specificity is not delineated considering that the comparison group was healthy controls and not patients suspected of a mitochondrial disease with subsequent genetic and or functional indication of an unrelated disease

When the patient groups are sub-divided into their affected systems, however, the sensitivity and strength of the predictions becomes variable. Isolated CI and CIV defects were the most likely groups to show impaired activity in their respective systems, with 71% and 100% of cases, respectively, being characterized as “very likely” or “likely”.

We did not perform the relevant CV spectrophotometric assay, so unsurprisingly, the sole patient in this category was spuriously characterized as “unlikely” by our scoring method.

One patient with a mitochondrial structural defect was categorized as “unlikely”, suggesting no secondary OXPHOS defect.

Mutations in the mtDNA system (mtDNA mutations, and mtDNA transcription and translation defects) were detected with 68% sensitivity. Contained within this number though is that only 47% of these patients were either characterized as either “likely” or “very likely”, with a further 21% only as “possible”. Included in the “unlikely” group are the two patients with large mtDNA mutations, a group which is anecdotally (unpublished) only diagnosed through a muscle biopsy. We speculate that enzymology has limited sensitivity in detecting these patients, as combinations of perhaps more minor deficiencies may become obscured when assessed as isolated complex measurements. Combined OXPHOS mutations also frequently have their roots in an mtDNA mutation. Therefore, heteroplasmy levels in fibroblasts, as compared to the affected tissue, is a further complicating factor.

Enzymological analysis of CI–IV activity in TCA cycle deficient patients is only expected to identify patients with secondary OXPHOS defects. To this end, 50% of patients in this group attracted the label “possible”. While enzymological methods are available for testing PDHC activity, which were not performed here, this data serves to highlight that the enzymological method is only as powerful as the depth of the analysis performed. We hypothesized that this limitation could be overcome by oxygraphy.

### 2.3. Oxygraphy

The oxygraphy method measures oxygen saturation in solution and therefore allows for the derivation of the rate of oxygen consumption in a sample (Figure 3a). Accordingly, the assay measures the rate of CIV activity (minus non-mitochondrial respiration measured as the antimycin A rate), which is constrained by the substrates and inhibitors that are injected into the chamber. By this means, indirect measurements of the TCA cycle, and OXPHOS CI–V and the glycerophosphate dehydrogenase complex can be deduced. The assay therefore has the potential to provide a much broader insight into mitochondrial activity in a single assay than enzymology, which requires additional testing for each new complex of interest. For example, substrates including pyruvate must be metabolized through the PDHC complex, glutamate through α-ketoglutarate dehydrogenase complex, and several of the TCA cycle enzymes require CoA as a co-factor, as is assumed to be impaired in the case in SLC25A42 deficiency (mitochondrial CoA transporter).

An example of a data recording is shown in Figure 3a, including the points from where the primary measures are derived. Rates of oxygen consumption under the conditions described, and derived measures, are provided in Figure 3b,c. The calculated results, as well as a disease prediction (see Materials and Methods), are presented in Table 3. A definition and brief guide to data interpretation is provided in Table 4.

Taken together, mitochondrial dysfunction was suspected in 79% of cases (“possible”, “likely”, or “very likely”). 92% of controls were identified as “unlikely”, and one control (8% of the control cohort) was identified as “possible”. All control values followed a normal distribution as tested by both the Shapiro–Wilk and Kolmogorov–Smirnov tests in Prism 8, where p ≥ 0.05 was considered to follow a normal distribution.

It should be noted that technical replicates from controls and patients showed a significant spread (Appendix A for control spread only), with the exception of the glutamate addition and Q-point ratios. Numerous methods were trialed to reduce this spread, including by normalizing the data to the resting rate of oxygen consumption, stringent plating methods including controlling for passage number, and normalization to protein content (data not shown). Unfortunately, none of these methods appeared to significantly reduce the spread, which was also observed when the same preparation of cells was simultaneously injected into the two different chambers of the oxygraph. Normalization to CS activity was another potential mechanism to normalize the data. However, considering that CS rates of activity as measured by enzymology (Figure 2 and Table 2) fell within a relatively narrow range, and that the variation observed when the same cellular preparation was injected into two chambers, we did not expect this would reduce the spread, and accordingly did not further examine this. Reducing the technical variation was surmounted in our hands by performing three technical replicates per sample.

At the sub-level of isolated CI mutations, 100% of cases were predicted as “very likely” or “likely”. Dysfunction manifested in several measures linked to CI activity, but most notably in the resting rate, and in the uncoupling increase. Somewhat surprisingly, dysfunction also frequently manifested as impaired isolated CIV activity, thus clouding our ability to judge the biochemical basis of this defect without enzymological testing or genetics. While we cannot provide an explanation for this phenomenon, we speculate that since mitochondrial volume appears to be relatively normal in these lines (see CS measures Figure 2 and Table 2), it may be linked to super-complex formation [19]. This is considered given that the OXPHOS system is intact in oxygraphy, unlike its dissociated state in enzymology. Accordingly, impaired CI activity in a super complex, such as in the respirasome (CI/III_2_/IV), may cause secondary dysfunction in associated members of the super-complex. Glycerophosphate linked respiration was measured in anticipation that in the case of impaired CI activity, these cells may display enhanced glycerophosphate metabolism owing to its ability to bypass CI. This was, however, not the case. Similarly, fatty acid linked oxidation linked metabolism was tested with octanoylcarnitine, however neither controls nor PMD fibroblasts were found to be responsive (data not shown), perhaps owing to the saturation of the respiratory chain with existing CI and CII linked substrates pre-octanoylcarnitine injection.

Isolated CIV mutations were also indicated as “very likely” in 100% of cases. Not surprisingly, given the nature of oxygraphy measurements being directly linked to CIV activity, all raw rates were significantly impaired. The patients were also distinguished by a low Q-point ratio, consistent with coenzyme Q reduction by CIV being limiting.

The one CV patient in our cohort was also diagnosed by oxygraphy as “likely”. Underlying this diagnosis, though, was impaired uncoupled activity, where coupled activity was relatively normal and the CCR and uncoupling increase was low. This is counter to our expectations for a CV patient, and unlike CIV dysfunction in the case of isolated CI deficiency described above, cannot be readily explained by super-complex formation.

Patient cell lines with mutations in their mtDNA or mtDNA transcription and translation machinery were considered suggestive of PMD in 67% of cases. Only 61% of these, however, were considered “very likely” or “likely”, a relatively poor result. This includes two patients with large mitochondrial DNA deletions who were labelled as “unlikely”. Given the combined OXPHOS nature of these defects, dysfunction was indicated in many measures and ratios consistent with systemic disruption of the OXPHOS system.

The one patient with a mitochondrial structural defect was diagnosed as “possible”, as indicated most clearly by low coupled respiratory activity.

TCA cycle patients were indicated in 75% of cases as “very likely”. The underlying basis for this diagnosis varied significantly depending on the genetic change of the specific PMD patient. For PDHC patients, the standout measure was glutamate addition, consistent with an intact OXPHOS system which is starved if fueled from pyruvate, and operational as fueled by glutamine, which bypasses PDHC. For the one SLC25A42 patient which we labelled as “very likely”, the outstanding measure was the Q-point ratio. This is consistent with mitochondrial Coenzyme A depletion, which is bypassed by CII metabolism with succinate.

Overall then, oxygraphy provided a relatively high sensitivity and strength of diagnosis. However, with the exceptions of PDHC mutations, and to a lesser extent CIV (SURF1 mutations), there is little correlating signature between the oxygraphy profile and the cells’ underlying mutation.

### 2.4. Oxygraphy versus Enzymology

Oxygraphy outperforms enzymology for overall sensitivity and provides much stronger discriminatory power (far more likely to suggest “very likely” or “likely” by our scoring methods) than enzymology in PMD patient fibroblasts (Figure 4a,b). In particular, oxygraphy captured the mtDNA group much more clearly than enzymology (Figure 2 and Figure 3, Table 2 and Table 3), notably including the common POLG mutation which was poorly or not at all indicated by enzymology (Table 2 and Table 3). Neither oxygraphy nor enzymology, however, captured large mtDNA deletions. Oxygraphy also detected TCA cycle abnormalities with 75% sensitivity, a group that was not captured by respiratory chain CI–IV enzymological analysis as described here. The higher sensitivity of the oxygraphy method compared to enzymological method also appears to be reflected through tighter clustering of controls in a PCA analysis and heat map view of the measured and derived values (Appendix A).

When dysfunction is manifested by enzymology however, the profile much more clearly indicated the underlying molecular defect. Specifically, isolated CI, CIV, or combined OXPHOS defects were characterized as such correctly, unlike oxygraphy, which tended to only indicate that dysfunction was present, and not which system was affected.

The two techniques however did not always make overlapping diagnostic predictions. The sensitivity of the two techniques together, when the highest prediction is taken, rises to a very generous 88% (Figure 4a,b).

Neither oxygraphy or enzymology correlated closely with disease biometrics, including the age of onset, age of death, or clinical disease prediction score (Appendix A). This was true where all patients were compared as a group or subdivided into their genetic basis in a mtDNA or nDNA mutation, or further as compared to the individual test scores, or as a combined score. Nonetheless, these correlations may be restricted by the small sample size, notably when correlating against age of death.

## 3. Discussion

The complex clinical presentations and molecular basis of PMD, and their phenotypic and biochemical overlap with SMD, is an ongoing challenge in their diagnosis, and many patients experience a long and conflicting diagnostic odyssey [4]. NGS technologies have revolutionized the diagnostic odyssey for many of these patients, which, when combined with targeted genetic investigations, provide a diagnostic result in an estimated 50% of patients [20,21]. Genetic approaches still commonly identify variants of unknown significance in known or novel disease genes or fail to identify or suggest a disease-causing gene completely (estimated 50% of patients who have access to such advanced NGS tools). It is in this context of ambiguous genetic findings that biochemical tools are required such that they can be considered together with the available genetic findings (such as variants of unknown significance or suspected new disease genes) and clinical phenotype when assessing the likelihood of diagnosing a PMD. While traditionally this biochemical niche has been filled by enzymology testing of the OXPHOS system, we inquired if the oxygraphy method may also have clinical utility in diagnosing PMD. The oxygraphy technique was chosen as unlike enzymology, the assay is relatively quick to perform, and provides a broader insight into mitochondrial function than the OXPHOS system alone.

We performed a retrospective biochemical screen of 34 genetically defined PMD disease patient fibroblast cell lines by enzymology and oxygraphy to assess the diagnostic capabilities of both techniques. Fibroblasts were chosen, as they are available from a minimally invasive biopsy and can be readily maintained in culture, as compared to the preferred liver and muscle for enzymology, which are invasively obtained. It should be further considered that tissue from the commonly affected organ in PMD, such as the central nervous system, cannot be directly tested at all. While we acknowledge that skin disease is not a common presentation in PMD (most typically heart, muscle, liver, kidney, or CNS dysfunction), fibroblasts are utilized here in a diagnostic context to indicate the presence of a PMD or not, and therefore do not provide insights into the organ specific nature of the disease. Similarly, we acknowledge that our PMD fibroblasts are maintained in a 2-D culture environment, in which oxygen saturation and mitochondrial function are altered from their physiological state. While these variables are equally applicable for control and PMD fibroblasts, we cannot discount effects that may disproportionally favor mitochondrial function in PMD cells and so reduce the diagnostic sensitivity of both the enzymology and oxygraphy methods. For example, oxygen partial pressure is significantly higher in an in vitro cell culture setting (19%) compared to 2–5% in in vivo tissues [22]. Accordingly, in the situation that PMD arises due to reduced affinity for oxygen, higher oxygen partial pressures in the in vitro culture system may rescue (or partially) the defect without influencing control cells function, thus masking underlying dysfunction in cultured PMD fibroblasts.

It is also critical to note that our results, as defined though our diagnostic scoring matrix, are not intended to be applied as a definitive guide as to the presence of a PMD, but must be considered together with the clinical presentation of the patient, and any genetic results (see Figure 5 for suggested interpretation of enzymology and oxygraphy results in the context of variants of uncertain genetic significance).

Here we find that while oxygraphy outperforms enzymology for both sensitivity and predictive strength (“unlikely” to “very likely”), the enzymological method provides more in-depth biochemical information about the nature of the defect (e.g., isolated or combined OXPHOS dysfunction; Figure 5). In favor of the oxygraphy method, however, is implicit in the intact nature of the cells in the assay, and the reliance on multiple mitochondrial functions for respiratory activity. In this regard, oxygraphy has the capacity to detect defects in and closely associated with the TCA cycle, which are otherwise not captured by enzymology. By translation however, while not tested in this analysis, oxygraphy may also be more likely to detect mitochondrial dysfunction in SMD patients than by direct testing of the OXPHOS system using enzymology, thus causing the specificity of diagnosis made by the oxygraphy method to be diminished relative to enzymology. Despite this potential limitation, oxygraphy can be performed in only three hours for two cell lines (including cell culture), or less if multiple machines are available. While we recommend that three technical replicates are performed for accurate measurements by the oxygraphy method, this is still minimal effort in comparison to the enzymological method, which can take days to analyze the suite of enzymes necessary to cover the full array of possibly affected enzymes.

Ultimately, we conclude that while oxygraphy is a powerful new tool for the biochemical diagnosis of PMD, the two techniques are complementary, and should ideally be deployed synergistically (Figure 5). In this regard, the sensitivity rose to 88% when both techniques were applied together. It should be questioned then if an invasive muscle, heart, or liver biopsy is warranted before performing these tests first in fibroblasts.

Finally, we reinforce that both biochemical tests must be assessed in conjunction with clinical features, other available biochemical data (such as COX staining, BN-PAGE, or measurements of ATP synthesis capacity), and genetic data in determining the likeliness of a PMD, and as a standalone these tests do not offer definitive proof of a PMD.

## 4. Materials and Methods

### 4.1. Ethics

Analysis of fibroblasts at the UZ Leuven Hospital were in accordance with ethics application number S60206 (retrospective metabolic analysis of archived fibroblasts).

### 4.2. Cell Culture

Control and PMD patient fibroblasts at less than passage 15 were maintained in low glucose DMEM medium (ThermoFisher) with 5.5 mM glucose (closest approximation to physiological glucose concentration in plasma) and 2 mM glutamine supplemented with 10% foetal calf serum at 37 °C with 5% CO_2_ in a humidified incubator. Cells for both enzymology and oxygraphy were cultured to a density of 50–80% before being harvested by trypsinization, washed twice in PBS, and either re-suspended to 20 million cells/mL for oxygraphy, or snap frozen in a dry ice ethanol slurry for enzymology.

### 4.3. Enzymology

Enzymological methods were validated against historical clinical results generated from multiple centers (UZ Ghent, Belgium, Radboud Nijmegen hospital and the Academic Medical Centre Amsterdam, The Netherlands) in the course of routine diagnostics performed on these patients, which, when available in fibroblasts, showed close concordance in disease prediction to our own testing. Enzymology was performed as previously described [10]. Briefly, cells cultured and harvested as above from 4 X T175 flasks per cell line were stored at −80 °C before resuspension in 1 mL ice cold Mega Fb buffer (250 mM sucrose, 2 mM HEPES, 0.1 mM EGTA, pH 7.4) and homogenized on an ice slurry in a Teflon-glass Wheaton homogenizer by application of 20 strokes driven by a Glas-Col High Speed Homogenizer variable speed bench top drill at 1800 rpm. Cell debris was pelleted in a low speed spin (600 g, 10 min, 4 °C, no break), and the pellet was re-dissolved in 800 µL Mega Fb buffer, debris pelleted as before, and the two tubes of supernatant were combined. The combined supernatant was then centrifuged (14,400 g, 10 min, 4 °C, no break). The supernatant was discarded, and the pellet was resuspended in 400 µL ice cold Mega Fb buffer. 75 µL was removed for CIII measurement. The remaining supernatant was centrifuged (14,400 g, 10 min, 4 °C, no break), and the pellet was resuspended in 1 mL of Hypotonic buffer (25 mM potassium phosphate, pH 7.2, 5 mM MgCl_2_), pelleted (14,400 g, 10 min, 4 °C, no break), and finally the pellet was resuspended in 400 µL of Hypotonic buffer for analysis of CI, II, and IV activity. All preparations for enzymological analysis were finally subjected to three rounds of freeze thaw cycles in a dry ice and ethanol slurry, before being stored at −80 °C prior to analysis.

Enzyme assays were performed in technical duplicates from the same preparation of cells on individual complexes of the respiratory chain (CI–CIV), and citrate synthase (CS) by spectrophotometric methods on a Carey 3000 spectrophotometer (Agilent), as previously described [10].

### 4.4. Oxygraphy

Oxygraphy was performed as previously described [23,24] with a technical n of ≥ 3. Briefly, cells cultured and harvested as above from a T175 were resuspended in Miro5 buffer to 20 million cells/mL and 100 μL (2 million cells) were injected per chamber of the oxygraph, with subsequent injections of the following compounds to the final concentration of: digitonin, (Merck) 7.5 µg/mL as determined by a digitonin titration; pyruvate, 5 mM; malate, 0.5 mM; ADP (Calbiochem), 1 mM; glutamate, 10 mM; succinate, 10 mM; carbonyl cyanide m-chlorophenyl hydrazone (CCCP), Δ0.5 µM until maximum respiration reached; rotenone, 75 nM; glycerophosphate, 10 mM; antimycin A, 250 nM; ascorbate sodium salt (Merck), 2 mM; N,N,N′,N′-Tetramethyl-p-phenylenediamine dihydrochloride (TMPD), 0.5 mM; sodium azide, 200 mM. Oxygen saturation was maintained at ≥ 20 µM throughout the experiment, and at ≥ 180–200 µM before the measurement of isolated complex IV (CIV) activity.

### 4.5. Reagents

All reagents were from Sigma unless otherwise specified.

### 4.6. Clinical Diagnostic Prediction Scoring

Retrospective clinical diagnostic scoring of genetically defined PMD patients was performed as previously described [25].

### 4.7. Diagnostic Prediction

Dysfunction indicating disease was assessed as either being “very likely”, “likely”, “possible”, or “unlikely” using Z scores falling below median control values, with the exceptions of the glutamate addition and glycerophosphate additions (higher than control median), and the Q-point (absolute Z score was assessed) measures. Disease was predicted using the Z score where either an individual measure or the sum of the positive Z scores met a threshold, as described in Table 5. Sensitivity was defined as the percentage of patients not labelled as “unlikely”. Z score thresholds were chosen for optimal separation of controls and PMD fibroblasts, with a step wise increase in Z scores, indicating to us an increased likelihood of mitochondrial dysfunction and thus likeliness of a PMD.

### 4.8. Statistics

Unless otherwise described, error bars describe the calculated 1.25th and 98.75th percentiles of the reference range, and green shading shows the range of the data in control patients. The word “significant” is only used here to refer to perceived importance and is not used in reference to any statistical tests.

## Figures and Tables

**Figure 1 metabolites-09-00220-f001:**
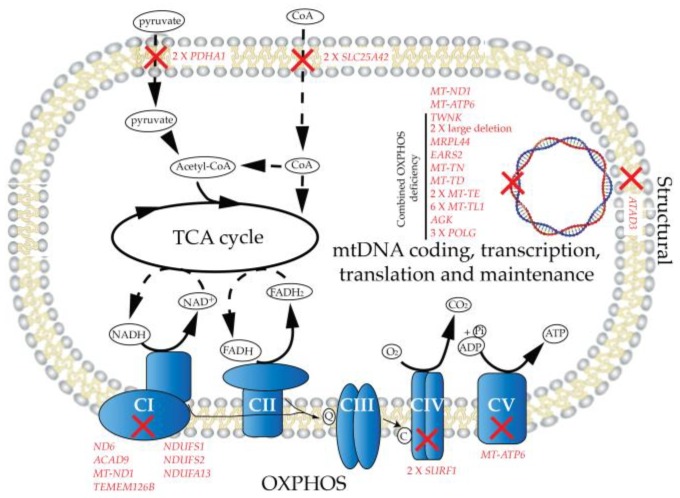
Mutational mapping. Genetic changes (red) of primary mitochondrial disease (PMD) patients reported in this study mapping to the mitochondrial systems of: primary OXPHOS, TCA cycle connections, the mtDNA system, and structural components. Abbreviations: OXPHOS CI–V, oxidative phosphorylation complexes I–V; TCA, tricarboxylic acid cycle.

**Figure 2 metabolites-09-00220-f002:**
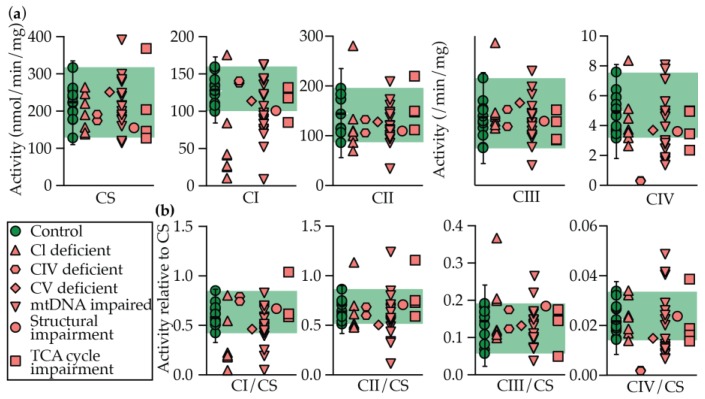
Enzymology in control and PMD patient fibroblast cell lines. Control and PMD patient fibroblasts were measured CS and respiratory chain complex I–IV (CI–IV) activity by spectrophotometric methods. Results are presented as either (**a**) raw rates, or (**b**) relative to CS activity. Median is displayed for controls with error bars showing the 1.25th and 98.75th percentiles of the reference range, and the green shading region shows the range. Each data point represents the average of each patient or control from ≥ 2 technical replicates Abbreviations: CI–IV, respiratory chain complexes I–IV; CS, citrate synthase.

**Figure 3 metabolites-09-00220-f003:**
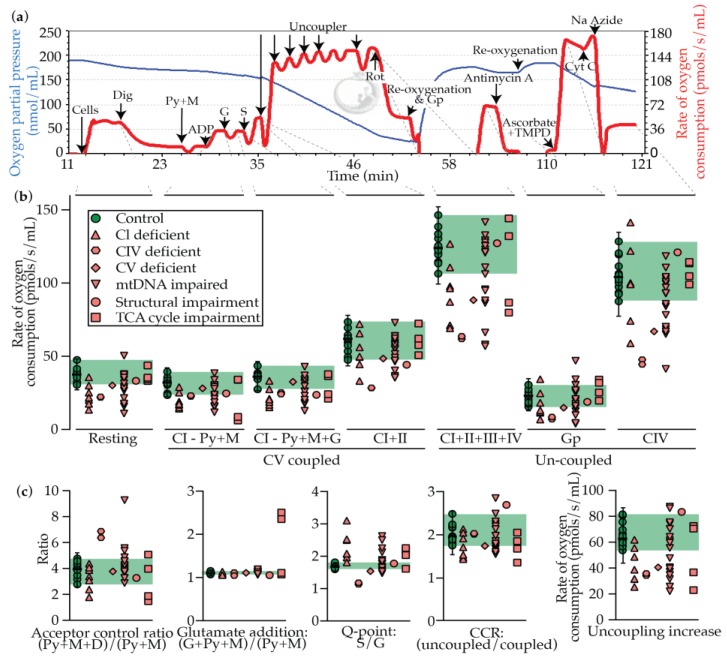
Oxygraphy testing in control and PMD patient fibroblast cells. (**a**) Representative trace from control fibroblasts, with substrates and inhibitors injected as described. Blue line indicates the oxygen partial pressure in the chamber, and the red line indicates the inverted rate of change of the blue line (rate of oxygen consumption). (**b**) Resting, coupled, and uncoupled rates of respiration. (**c**) Ratios and calculated values from data in panel (**b**). Median is displayed for controls with error bars showing the 1.25th and 98.75th percentiles of the reference range, and the green shading region shows the range. Each data point represents the average of each patient or control from ≥ 3 technical replicates. Abbreviations: As, ascorbate; CI–IV, respiratory chain complexes I–IV; CCR, coupling control ratio (maximal uncoupled activity over maximal coupled activity; Cyt C, cytochrome C; Dig, digitonin; Gp, Glycerophosphate; G, glutamate; M, malate; Py, pyruvate; Rot, rotenone; S, succinate.

**Figure 4 metabolites-09-00220-f004:**
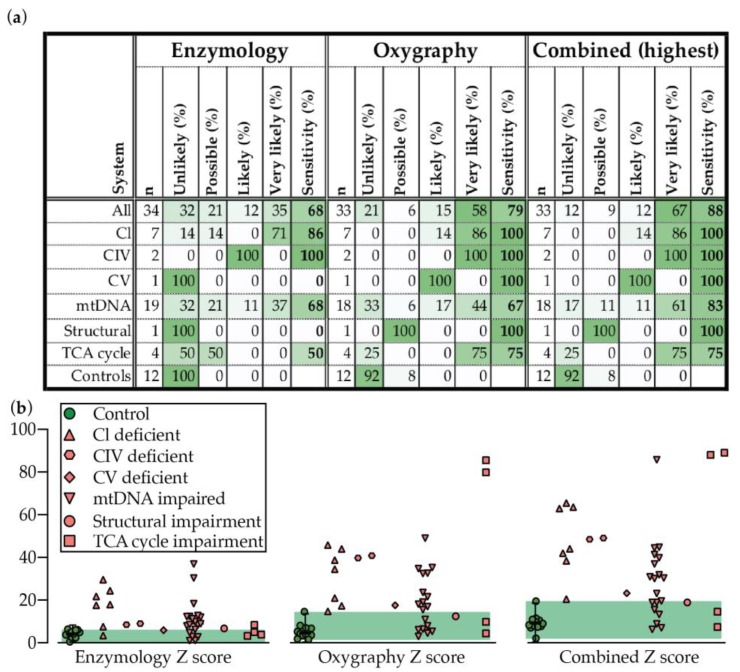
Disease prediction and sensitivity for enzymology, oxygraphy, or combined methods for detecting PMD as presented (**a**) in a table, or (**b**) through the plotting of the combined positive Z scores for each test and combined value. Median is displayed for controls with error bars showing the 1.25th and 98.75th percentiles of the reference range, and the green shading region shows the range. Abbreviations: CI–CIV, respiratory chain complexes I–IV.

**Figure 5 metabolites-09-00220-f005:**
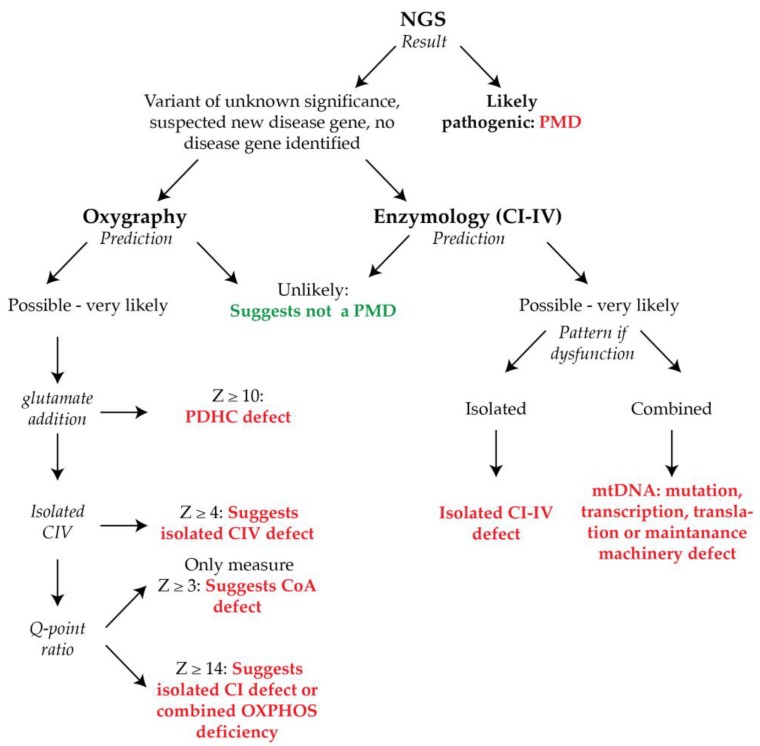
Diagnostic guide to interpreting enzymology and oxygraphy results in the absence of a definitive genetic result. Next generation sequencing or targeted genetic mitochondrial panels can provide a definitive diagnosis of mitochondrial disease and should alleviate the need for further biochemical testing. In cases of non-definitive diagnostic results, biochemical testing is required. In this regard, oxygraphy and enzymology would ideally be deployed synergistically for the optimal chance of detecting dysfunction in fibroblasts, and for the identification of the type of underlying defect.

**Table 1 metabolites-09-00220-t001:** PMD patient characteristics. Abbreviations: CI–CV, OXPHOS complexes I–V; CDG, congenital disorder of glycosylation; CKD, chronic kidney disease; Com, combined; DM, diabetes mellitus; F, female; GDD, global developmental delay; HCM, hypertrophic cardiomyopathy; LTBL, leukocencephalopathy with thalamus and brainstem involvement and high lactate +; M, male; MIU, mors in utero; NALF, neonatal acute liver failure; ND, no data; PEO(+), progressive external ophtalmoplegia (with accompanying symptoms).

Identifier #	Gene	Mutation	System	Gender	Disease	Symptoms	Age of Onset (years)	Current Age (years)	Age at Death (years)	Diagnostic Score (/8, see Appendix A)
45	*TMEM126B*	c.401het_delA (p.134N>IIefs*2), c.635G>T (p.212G>V)	Cl	M		Myopathy	11	40		6
30	*ACAD9*	c.976G>C (p.326A>T), c.1552C> T (p.518R>C)	Cl	F	Hypertrophic cardiomyopathy	HCM	2	8		6
33	*MT-ND1*	mtDNA.3481G>A (p.59E>K)	Cl	F		GDD, cardiomyopathy, lactic acidosis	0		2	8
48	*ND6*	mtDNA.14487T>C (p.63E*)	Cl	M		Acute vision loss, progressive myoclonic epilepsy with extrapyramidal syndrome and psychosis	19	43		7
2737	*NDUFS1*	c.1057G>C (p.353A>P), c.420+2T>C (splice site mutation)	Cl	M	Leigh syndrome	GDD, neurocognitive regression	2	3		8
2736	*NDUFS2*	Homozygous: c.1336G>A (p.446D>N)	Cl	F	Leigh syndrome	Necrotic encephalopathy after vaccination	0		0	6
2497	*NDUFA13 AND PGM1*	NDUFA13, homozygous: c.170G>A(p.57R>H), PGM1, homozygous: c.1108A>T (p.370K*)	CI	F	Leigh syndrome/CDG	Deafness, GDD, spastic dystonic quadriplegia, epilepsy	0.5		17	8
52	*SURF1*	c.312del10 insAT (p.fs*), c.544 GT>CA (p.182V>H)	CIV	F	Leigh syndrome	Ataxia, myopathy, respiratory insufficiency	1		8	8
55	*SURF1*	c.845-856del (p.282S>Cfs*), c.870insA (p.292K>E)	CIV	F	Leigh syndrome	Ataxia, dystrophy, FTT, renal tubular acidosis	2		3	8
2264	*MT-ATP6*	mtDNA.8993T>G (p.156L>R)	CV	M	Infantile NARP	GDD, ataxia, epilepsy, dystrophy	1	19		8
47	*AGK*	c.409C>T (p.137R>X), c.1131+5G>A (splice site exon 15)	Com OXPHOS	M	Sengers syndrome	Congenital cataract, HCM, myopathy	0		23	8
34	*EARS2*	c.286G>A (p.96Q>K), c.500G>A (p.167C>Y)	Com OXPHOS	M	LTBL	GDD	1	12		7
43	*MRPL44*	ND	Com OXPHOS	M		Myopathy, cardiomyopathy, encephalopathy with epilepsy	1	33		5
42	Large mtDNA deletion	mtDNA.12113_14421del2309	Com OXPHOS	F	Kearns-sayre	Bilateral ptosis, scoliosis, myopathy, ophtalmoplegia	16	63		2
41	Large mtDNA deletion	mtDNA.8937_14422del	Com OXPHOS	F	PEO+	Ptosis, PEO, dysphagia, myopathy	12	63		6
50	*MT-TD*	mtDNA.7526A>G	Com OXPHOS	F		Myopathy, migraine	9	36		6
36	*MT-TE*	mtDNA.14674T>G	Com OXPHOS	F		GDD, metabolic decompensations, CKD	0	15		4
57	*MT-TE*	mtDNA.14709T>C	Com OXPHOS	F		Hypotonia, GDD, DM	0	14		7
58	*MT-TL1*	mtDNA.3291T>C	Com OXPHOS	F		Myopathy, respiratory failure (on ventilation), DM, CKD , HCM	41	73		4
123	*MT-TL1*	mtDNA.3261A>G	Com OXPHOS	F		Myopathy, exercice intolerance, lactic acidosis, sudden death during respiratory infection at home	1		33	7
53	*MT-TL1*	mtDNA.3243A>G	Com OXPHOS	M	MELAS	Cardiopathy, DM, deafness, frontal syndrome, myopathy, ophthalmoplegia	41	61		8
54	*MT-TL1*	mtDNA.3243A>G	Com OXPHOS	F	MELAS	Exercice intolerance, lactic acidosis, epilepsy	10	35		6
72	*MT-TL1*	mtDNA.3243A>G	Com OXPHOS	M	MELAS	DM, epilepsy, pseudo-strokes, deafness	30		42	8
40	*MT-TL1*	mtDNA.3243A>G	Com OXPHOS	F	MELAS	DM, deafness, HCM, weight loss, CKD, biliary cysts	40		72	4
51	*MT-TN*	mtDNA.5728A>G	Com OXPHOS	M		Growth hormone deficiency, CKD, GDD, epilepsy, myopathy	2.3		17	8
124	*TWNK*	Heterozygous c.1358G>C (p.453R>P), WT	Com OXPHOS	M	PEO+	Myopathy with external ophtalmoplegia	42	55		3
35	*POLG*	c.1402A>G(p.468N>D), WT	Com OXPHOS	F	Alpers	Liver fibrosis, ataxia, spastic hemiparesis	38	58		7
38	*POLG*	c.1399G>A (p.467A>T), c.2542G>A (P.848G>S)	Com OXPHOS	F	Alpers	NALF, refractory epilepsy	1		1.5	7
120	*POLG*	c.1252T>G (p.418C>G), WT	Com OXPHOS	M		Myopathy	56	81		3
59	*ATAD3*	c.1582C>T (p.528R>W), WT	Structural	F		GDD, Spastic dystonic quadriplegia, morphea	1	7		6
2130	*PDHA1*	c.904C>T (302R>C), WT	TCA cycle	F		GDD, spastic dystonic quadriplegia, epilepsy	0.6	40		6
31	*PDHA1*	c.523G>A (p.175A>T), WT	TCA cycle	F		Deafness, infantile spasms, GDD	0	6		8
128	*SLC25A42*	Homozygous c.309C>G (p.103Y>X)	TCA cycle	M		Myopathy, acidocetosis	9	28		7
2738	*SLC25A42*	Homozygous c.871A>G (p. 291N>D)	TCA cycle	M		GDD, lactic acidosis, severe spastic quadriplegia, dysarthria, severe kyphosis, epilepsy	<5	30		6

**Table 2 metabolites-09-00220-t002:** Enzymological testing results and disease predictions. Enzymology values for individual PMD patient cell lines as grouped based on the system effected. Disease prediction was assigned using Z scores, as described in the Materials and Methods. Abbreviations: CI–IV, respiratory chain complexes I–IV; CS, citrate synthase; Str, structural.

Identifier #	Gene	System	Rates of Enzymes Activity and Ratios to CS	Z Scores	Disease prediction
CS (nmol/min/mg)	CI (⎢nmol/min/mg⎢)	Rotenone sens (%)	CII (⎢nmol/min/mg⎢)	CIII (⎢/min/mg⎢)	CIV (⎢/min/mg⎢)	CI/CS	CII/CS	CIII/CS	CIV/CS	CS (nmol/min/mg)	CI (⎢nmol/min/mg⎢)	Rotenone sens (%)	CII (⎢nmol/min/mg⎢)	CIII (⎢/min/mg⎢)	CIV (⎢/min/mg⎢)	CI/CS	CII/CS	CIII/CS	CIV/CS	Sum (+ Z only)
**45**	***TMEM126B***	Cl	143	27	42	69	29	2.6	0.19	0.48	0.20	0.018	1.7	5.2	8.4	1.9	−0.3	1.6	3.3	1.5	−1.3	0.5	24.2	Very likely
30	*ACAD9*	157	84	67	110	57	3.7	0.54	0.70	0.37	0.024	1.4	2.3	1.8	0.9	−3.8	0.8	0.3	−0.6	−4.7	−0.3	7.4	Possible
33	*MT-ND1*	139	25	39	86	28	4.5	0.18	0.62	0.20	0.032	1.7	5.3	9.1	1.5	−0.1	0.3	3.4	0.2	−1.1	−1.6	21.5	Very likely
48	*ND6*	221	176	80	108	26	5.1	0.80	0.49	0.12	0.023	0.1	−2.5	−1.6	0.9	0.1	−0.2	−1.9	1.5	0.6	−0.2	3.2	Unlikely
2737	*NDUFS1*	247	42	41	280	24	8.4	0.17	1.14	0.10	0.034	−0.5	4.4	8.6	−3.6	0.3	−2.5	3.5	-4.8	1.0	−1.9	17.7	Very likely
2736	*NDUFS2*	265	10	20	133	29	3.5	0.04	0.51	0.11	0.014	−0.8	6.1	14.2	0.2	−0.2	1.0	4.6	1.3	0.8	1.3	29.4	Very likely
2497	*NDUFA13 AND PGM1*	191	41	59	99	23	3.2	0.22	0.52	0.12	0.017	0.7	4.5	3.8	1.1	0.5	1.2	3.1	1.1	0.5	0.8	17.4	Very likely
52	*SURF1*	CIV	175	138	75	105	31	0.3	0.78	0.60	0.17	0.002	1.0	−0.5	−0.4	1.0	−0.5	3.3	−1.8	0.4	−0.6	3.1	8.8	Likely
55	*SURF1*	192	141	77	132	24	0.3	0.73	0.69	0.12	0.001	0.7	−0.7	−0.8	0.3	0.4	3.3	−1.4	−0.5	0.5	3.2	8.3	Likely
2264	*MT-ATP6*	CV	252	114	75	127	33	3.7	0.45	0.51	0.13	0.015	−0.6	0.7	−0.3	0.4	−0.8	0.9	1.1	1.3	0.3	1.1	5.7	Unlikely
47	*AGK*	mtDNA	285	52	44	33	20	1.8	0.18	0.12	0.07	0.006	−1.2	3.9	7.9	2.8	0.9	2.2	3.4	5.1	1.6	2.4	30.1	Very likely
34	*EARS2*	200	98	75	107	29	2.4	0.49	0.53	0.15	0.012	0.5	1.6	−0.4	0.9	−0.3	1.8	0.7	1.0	0.0	1.5	8.0	Possible
43	*MRPL44*	186	88	68	101	25	1.6	0.47	0.54	0.13	0.009	0.8	2.1	1.5	1.1	0.3	2.4	0.9	0.9	0.2	2.0	12.1	Very likely
42	Large mtDNA deletion	177	121	74	120	24	7.1	0.68	0.68	0.14	0.040	1.0	0.4	−0.2	0.6	0.4	−1.6	−0.9	−0.4	0.2	−2.9	2.5	Unlikely
41	Large mtDNA deletion	280	162	79	161	37	8.1	0.58	0.57	0.13	0.029	−1.1	−1.8	−1.4	−0.5	−1.2	−2.3	0.0	0.6	0.3	−1.1	0.9	Unlikely
50	*MT-TD*	215	83	65	143	32	2.4	0.39	0.67	0.15	0.011	0.2	2.3	2.2	0.0	−0.6	1.8	1.6	−0.3	−0.1	1.6	9.7	Likely
36	*MT-TE*	241	136	72	172	40	4.8	0.56	0.71	0.16	0.020	−0.4	−0.4	0.4	−0.8	−1.6	0.1	0.1	−0.7	−0.4	0.3	0.8	Unlikely
57	*MT-TE*	249	107	71	85	25	2.7	0.43	0.34	0.10	0.011	−0.5	1.1	0.7	1.5	0.2	1.6	1.3	2.9	0.9	1.7	11.8	Very likely
58	*MT-TL1*	300	142	76	147	22	3.0	0.47	0.49	0.07	0.010	−1.5	−0.7	−0.5	−0.1	0.6	1.4	0.9	1.4	1.5	1.8	7.7	Possible
123	*MT-TL1*	160	74	61	136	15	7.8	0.46	0.85	0.10	0.049	1.3	2.8	3.3	0.2	1.4	−2.1	1.0	−2.1	1.0	−4.2	11.0	Likely
53	*MT-TL1*	115	76	65	143	25	3.0	0.66	1.24	0.22	0.026	2.2	2.7	2.2	0.0	0.2	1.4	−0.7	−5.8	−1.6	−0.7	8.7	Possible
54	*MT-TL1*	216	9	16	119	8	1.7	0.04	0.55	0.04	0.008	0.2	6.2	15.3	0.6	2.4	2.3	4.6	0.8	2.3	2.1	36.7	Very likely
72	*MT-TL1*	392	163	70	208	28	5.6	0.42	0.53	0.07	0.014	−3.4	−1.8	0.9	−1.7	−0.1	−0.5	1.4	1.0	1.5	1.2	6.0	Unlikely
40	*MT-TL1*	286	70	60	93	29	2.8	0.24	0.32	0.10	0.010	−1.3	3.0	3.6	1.3	−0.2	1.5	2.9	3.0	0.9	1.9	18.1	Very likely
51	*MT-TN*	179	90	70	100	20	1.9	0.50	0.56	0.11	0.010	0.9	2.0	1.0	1.1	0.9	2.2	0.6	0.8	0.7	1.8	12.0	Very likely
124	*TWNK*	214	94	73	94	24	1.3	0.44	0.44	0.11	0.006	0.2	1.7	0.1	1.3	0.4	2.6	1.2	2.0	0.7	2.4	12.5	Very likely
35	*POLG*	176	144	73	112	46	3.9	0.82	0.64	0.26	0.022	1.0	−0.8	0.2	0.8	−2.4	0.7	−2.1	0.0	−2.5	−0.1	2.7	Unlikely
38	*POLG*	191	123	84	110	19	4.6	0.64	0.58	0.10	0.024	0.7	0.2	−2.6	0.8	1.0	0.2	−0.6	0.6	0.9	−0.4	4.5	Unlikely
120	*POLG*	120	83	67	97	20	5.0	0.69	0.80	0.17	0.041	2.1	2.3	1.8	1.2	0.8	-0.1	−1.0	−1.6	−0.5	−3.1	8.3	Possible
59	*ATAD3*	Str	156	101	67	109	26	3.6	0.66	0.71	0.18	0.023	1.4	1.4	1.7	0.9	0.1	0.9	−0.7	−0.7	−0.8	−0.3	6.5	Unlikely
2130	*PDHA1*	TCA cycle	205	118	70	147	30	3.4	0.58	0.73	0.15	0.019	0.4	0.5	1.1	-0.1	-0.4	1.1	0.0	−0.9	0.0	0.5	3.6	Unlikely
31	*PDHA1*	147	85	71	111	26	2.3	0.58	0.75	0.18	0.016	1.6	2.2	0.7	0.8	0.2	1.8	0.0	−1.1	−0.6	0.9	8.2	Possible
128	*SLC25A42*	369	225	76	219	18	5.0	0.61	0.59	0.05	0.013	−3.0	−5.0	−0.7	−2.0	1.1	−0.1	−0.3	0.4	2.0	1.3	4.8	Possible
2738	*SLC25A42*	128	132	76	149	19	4.9	1.03	1.16	0.15	0.038	2.0	−0.2	−0.7	−0.2	1.0	0.0	−3.9	−5.1	0.0	−2.6	3.0	Unlikely
Median (+Z only)	Controls	224	128	74	143	27	4.9	0.58	0.64	0.14	0.022	0.4	0.4	0.4	0.4	0.4	0.4	0.3	0.3	0.6	0.3	4.1	100% unlikely
Min	129	100	67	86	15	3.1	0.42	0.52	0.06	0.014	−1.9	−1.7	−1.7	−1.4	−2.0	−2.0	−2.3	−2.2	−1.0	−1.9	0.3
**Max**	**318**	**160**	**80**	**195**	**43**	**7.6**	**0.84**	**0.87**	**0.19**	**0.034**	**1.9**	**1.4**	**1.7**	**1.5**	**1.5**	**1.3**	**1.4**	**1.2**	**1.8**	**1.2**	**6.1**
1/4 percentile	184	109	72	113	21	3.6	0.51	0.59	0.08	0.018	−0.6	−0.7	−0.9	−1.0	−0.5	−0.7	−0.8	−1.0	−0.5	−0.9	2.3
3/4 percentile	253	142	77	182	31	5.9	0.67	0.74	0.17	0.027	0.8	1.0	0.6	0.8	0.8	0.9	0.6	0.5	1.3	0.6	5.6
n	12	12	12	11	12	12	12	11	12	12	12	12	12	11	12	12	12	11	12	12	12
% Coefficient of variation	22	15	5	27	30	28	20	16	33	30		
Shapiro-Wilk test (p=)	0.98	0.95	0.98	0.89	0.96	0.94	0.96	0.95	0.91	0.95
Kolmogorov-Smirnov test (p=)	>0.1	>0.1	>0.1	>0.1	>0.1	>0.1	>0.1	>0.1	>0.1	>0.1

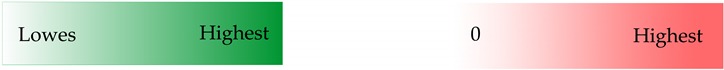

**Table 3 metabolites-09-00220-t003:** Oxygraphy testing results and disease predictions. Oxygraphy values for individual cell lines as grouped based on the system affected. Disease prediction was assigned using Z scores as described in the Materials and Methods. Abbreviations: As, ascorbate; CI–IV, OXPHOS complexes I–V; CS, citrate synthase; E, electron transfer state; Gp, glycerophosphate; G, glutamate; M, malate; P, phosphorylating state; Py, pyruvate; S, succinate; Str, structural.

Identifier #	Gene	System	Rates of Oxygen Consumption (pmols/s/mL) and Ratios	Z Scores	Disease prediction
Resting	P(CI - Py+M)	P(CI - Py+M+G)	P(CI+II)	E (CI+II+III+IV)	E(GP)	E(CIV)	Acceptor control ratio	Glutamate addition	Q-point: S/G	Coupling control ratio	Uncoupling increase	Resting	P(CI - Py+M)	P(CI - Py+M+G)	P(CI+II)	E (CI+II+III+IV)	E(GP)	E(CIV to As/TMPD)	Acceptor control ratio	Glutamate addition	Q-point: S/G	Coupling control ratio	Uncoupling increase	Sum (+ Z only)
45	*TMEM126B*	Cl	18	18	21	57	88	7	100	3.0	1.05	2.5	1.5	31	4.3	3.4	3.4	0.8	3.2	−3.4	0.4	1.5	1.4	14.9	1.9	3.3	38	Very likely
30.0	*ACAD9*	14	15	16	34	72	12	59	3.8	1.07	1.9	2.0	38	5.2	4.2	4.6	3.8	4.6	−2.4	3.6	0.2	0.7	5.0	−0.4	2.6	34	Very likely
33	*MT-ND1*	21	16	17	45	70	13	100	1.7	1.09	2.5	1.5	25	3.7	4.0	4.2	2.3	4.7	−2.2	0.4	3.8	0.0	14.5	2.2	4.0	44	Very likely
48	*ND6*	26	26	28	50	98	14	75	4.3	1.09	1.8	1.9	48	2.6	1.6	1.8	1.6	2.3	−2.0	2.4	0.7	0.1	2.4	0.2	1.5	17	Likely
2737	*NDUFS1*	22	17	19	66	97	25	101	3.3	1.09	3.1	1.4	31	3.4	3.6	3.8	−0.5	2.4	0.4	0.3	1.1	0.1	25.0	2.3	3.3	46	Very likely
2736	*NDUFS2*	36	30	34	73	127	35	142	4.0	1.12	2.1	1.7	55	0.3	0.7	0.5	−1.3	−0.3	2.4	−3.0	0.2	1.4	7.2	1.1	0.8	15	Very likely
2497	*NDUFA13 AND PGM1*	30	20	21	50	111	19	122	2.3	1.08	2.0	2.1	61	1.6	3.1	3.3	1.7	1.1	−1.0	−1.4	2.8	0.2	7.0	−0.8	0.1	21	Very likely
52	*SURF1*	CIV	22	23	26	29	62	7	48	6.3	1.10	1.1	2.0	34	3.3	2.2	2.3	4.4	5.3	−3.3	4.5	4.2	0.7	9.5	−0.3	3.0	39	Very likely
55	*SURF1*	23	24	25	29	64	9	45	6.8	1.03	1.1	2.0	35	3.2	2.1	2.6	4.4	5.2	−3.0	4.7	5.0	1.9	8.6	−0.4	2.9	41	Very likely
2264	*MT-ATP6*	CV	31	29	33	49	89	15	68	3.7	1.10	1.5	1.7	40	1.5	0.9	0.7	1.8	3.1	−1.7	2.9	0.3	0.4	2.4	1.0	2.4	17	Likely
47	*AGK*	mtDNA	ND																									
34	*EARS2*	35	39	43	64	124	31	93	5.5	1.10	1.4	1.9	59	0.7	−1.5	−1.6	−0.2	0.1	1.5	0.9	2.7	0.6	3.5	0.4	0.3	11	Possible
43	*MRPL44*	27	23	27	64	111	31	94	3.2	1.13	2.6	1.7	47	2.3	2.1	2.1	−0.2	1.1	1.6	0.9	1.3	1.5	16.6	1.3	1.6	32	Very likely
42	Large mtDNA deletion	36	29	32	58	121	27	119	3.6	1.09	1.7	2.0	61	0.3	0.8	0.9	0.6	0.3	0.6	−1.1	0.5	0.3	1.7	−0.1	0.1	6	Unlikely
41	Large mtDNA deletion	51	31	33	55	142	27	104	2.8	1.06	1.6	2.5	87	−2.9	0.4	0.8	1.0	−1.5	0.7	0.1	1.9	1.1	0.0	−2.4	−2.7	6	Unlikely
50	*MT-TD*	16	16	18	36	67	27	42	4.5	1.10	1.8	1.7	31	4.7	3.9	4.0	3.5	5.0	0.7	5.0	1.0	0.4	2.1	1.0	3.3	35	Very likely
36	*MT-TE*	37	30	34	56	130	26	100	4.1	1.14	1.6	2.3	74	0.1	0.6	0.5	0.8	−0.5	0.5	0.3	0.2	1.9	0.3	−1.4	−1.3	5	Unlikely
57	*MT-TE*	38	28	32	60	96	47	85	4.0	1.12	1.7	1.6	36	0.0	1.0	0.8	0.4	2.4	5.0	1.6	0.1	1.4	1.8	1.8	2.7	19	Likely
58	*MT-TL1*	38	30	34	73	128	35	102	3.6	1.13	2.1	1.7	55	−0.2	0.5	0.5	−1.4	−0.3	2.3	0.2	0.5	1.5	7.9	1.0	0.7	15	Very likely
123	*MT-TL1*	30	31	33	52	122	20	98	3.9	1.05	1.6	2.3	70	1.7	0.3	0.8	1.4	0.2	−0.7	0.6	0.0	1.3	1.3	−1.6	−0.8	8	Unlikely
53	*MT-TL1*	38	29	33	59	133	18	111	3.8	1.10	1.7	2.2	75	0.0	0.8	0.8	0.5	−0.8	−1.1	−0.5	0.3	0.4	1.6	−1.1	−1.4	4	Unlikely
54	*MT-TL1*	11	12	13	37	59	4	66	4.5	1.06	2.5	1.7	25	5.7	4.8	5.1	3.3	5.7	−4.0	3.1	0.9	0.7	14.5	1.0	3.9	49	Very likely
72	*MT-TL1*	18	17	19	47	87	13	79	4.8	1.11	2.2	1.8	40	4.3	3.8	3.8	2.1	3.2	−2.2	2.0	1.6	1.0	10.0	0.8	2.3	35	Very likely
40	*MT-TL1*	29	24	26	50	89	10	69	3.7	1.07	1.9	1.8	39	2.0	1.9	2.2	1.7	3.1	−2.7	2.8	0.4	0.4	3.7	0.7	2.5	21	Very likely
51	*MT-TN*	32	30	35	61	95	21	71	4.2	1.18	1.7	1.6	34	1.2	0.7	0.3	0.1	2.5	−0.4	2.7	0.4	3.4	1.8	1.8	3.0	18	Likely
124	*TWNK*	38	34	37	63	126	27	105	3.2	1.08	1.6	2.0	64	−0.1	−0.5	−0.3	0.0	−0.2	0.8	0.0	1.3	0.2	0.5	−0.1	−0.2	3	Unlikely
35	*POLG*	22	28	30	48	107	15	86	5.3	1.05	1.5	2.1	59	3.4	1.0	1.4	2.0	1.5	−1.7	1.4	2.4	1.2	2.2	−0.8	0.3	17	Likely
38	*POLG*	18	21	24	36	57	5	67	4.5	1.11	1.5	1.6	21	4.2	2.6	2.8	3.5	5.8	−3.8	2.9	0.9	0.9	2.6	1.6	4.3	32	Very likely
120	*POLG*	36	19	21	41	126	17	94	9.2	1.07	1.8	2.9	86	0.4	3.1	3.4	2.9	−0.1	−1.4	0.9	9.2	0.5	3.1	−4.1	−2.5	23	Very likely
59	*ATAD3*	Str	34	25	24	45	128	19	122	3.2	1.04	1.7	2.7	83	0.8	1.7	2.7	2.3	−0.3	−0.9	−1.4	1.2	1.6	1.8	−3.4	−2.3	12	Possible
2130	*PDHA1*	TCA cycle	34	7	22	51	87	35	114	1.4	2.50	2.1	1.7	36	0.9	6.1	3.3	1.5	3.2	2.3	−0.8	4.4	50.8	8.9	1.2	2.8	85	Very likely
31	*PDHA1*	35	9	25	58	80	20	100	1.8	2.35	2.2	1.4	22	0.6	5.5	2.6	0.6	3.8	-0.7	0.4	3.8	45.4	10.2	2.7	4.3	80	Very likely
128	*SLC25A42*	44	34	37	73	145	32	115	3.9	1.06	2.0	1.9	72	−1.4	−0.5	−0.2	−1.4	−1.8	1.8	−0.8	0.0	1.0	6.3	0.4	−1.1	10	Very likely
2738	*SLC25A42*	36	35	38	63	133	26	105	5.0	1.10	1.6	2.1	70	0.4	−0.5	−0.5	0.0	−0.7	0.5	−0.1	1.8	0.4	1.1	−0.5	−0.9	4	Unlikely
Median (+Z only)	Controls	38	32	36	63	124	23	104	3.9	1.09	1.6	2.0	62	0.3	0.4	0.4	0.5	0.3	0.4	0.3	0.7	0.8	0.7	0.2	0.2	5	92% unlikely, 8% possible
Min	31	24	28	48	107	16	88	2.7	1.06	1.6	1.8	53	−2.3	−1.8	−1.8	−1.5	−1.9	−1.6	−2.0	0.1	0.0	0.1	−2.4	−2.0	1
**Max**	**48**	**40**	**44**	**74**	**147**	**31**	**129**	**4.7**	**1.14**	**1.8**	**2.5**	**81**	**1.5**	**1.9**	**1.9**	**1.9**	**1.5**	**1.5**	**1.3**	**2.1**	**2.0**	**2.5**	**0.9**	**0.9**	**14**
1/4 percentile	34	31	35	55	116	19	95	3.5	1.07	1.6	1.9	58	−0.5	−0.4	−0.3	−0.3	−0.8	−0.9	−1.1	0.2	0.4	0.1	−1.2	−1.3	3
3/4 percentile	40	34	38	65	134	28	118	4.2	1.12	1.7	2.2	74	0.8	0.3	0.3	1.0	0.7	0.9	0.7	1.2	1.3	1.1	0.4	0.5	7
n	12	12	12	12	12	12	12	12	12	12	12	12	12	12	12	12	12	12	12	12	12	12	12	12	12
% Coefficient of variation	12	13	12	12	9	20	12	15	3	3	11	15	
Shapiro-Wilk test (p=)	0.56	0.23	0.16	0.80	1.00	0.82	0.78	0.60	0.16	0.32	0.34	0.18
Kolmogorov-Smirnov test (p=)	>0.1	0.05	0.05	>0.1	>0.1	>0.1	>0.1	>0.1	>0.1	>0.1	>0.1	>0.1

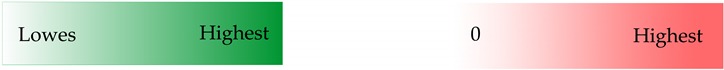

**Table 4 metabolites-09-00220-t004:** A brief guide to the interpretation of the oxygraphy data. Abbreviations: ΔΨ, mitochondrial membrane potential; CI–V, OXPHOS complexes I–V.

State	Description	Region	Interpretation
**Coupled respiration:** refers to the intact nature of the ΔΨ, and is thus limited by the proton motive force from CV (P)	Resting	Cells only	Unstimulated state with no substrates or inhibitors, and therefore could be influenced by any OXPHOS complex or TCA cycle impairment
	P(CI - Py+M)	As above + digitonin + pyruvate + malate + ADP	CI, III, IV, V and PDHC activity is limiting
	P(CI - Py+M+G)	As above + glutamate	CI, III, IV, V activity is limiting
	P(CI+II)	As above + succinate	CI, II, , III, IV, V activity is limiting
**Uncoupled respiration:** respiration refers to the state in which the ΔΨ is abolished, and is thus limited by the electron transport chain (E, CI-CIV)	E (CI+II+III+IV)	As above + uncoupler	Maximal uncoupled rate, which should therefore be limited by CI-IV activity
	E(GP)	(As above + glycerophosphate) - (as above)	Limited by glycerol-3-phosphate dehydrogenase (mGPDH)
	E(CIV)	(TMPD + Acorbate) - azide	Isolated rate of CIV activity
**Derived values**	Acceptor control ratio	$\frac{P(\mathrm{CI}\;-\;\mathrm{Py}+M)}{P(\mathrm{CI}\;-\;\mathrm{Py}+M)\;-\;\mathrm{ADP}}$	Influenced by the abundance of endogenous ADP, and is therefore an indicator of the charge ratio (ATP on ATP + ADP + AMP)
	Glutamate addition	P(CI - Py+M+G) P(CI - Py+M)	Indicates that pyruvate is limiting, pinpointing a PDHC deficiency
	Q-point: S/G	$\frac{P(\mathrm{CI}+\mathrm{II})}{P(\mathrm{CI}\;-\;\mathrm{Py}+M+G)}$	Limited by CI (high ratio) and CII (low ratio)
	Coupling control ratio	E (CI+II+III+IV) P(CI+II)	A large increase would be expected to indicate impaired CV activity, while a smaller increase should indicate impaired CI-CIV activity
	Uncoupling increase	E (CI+II+III+IV) - P(CI+II)	As above

**Table 5 metabolites-09-00220-t005:** Diagnostic prediction Z score thresholds.

		Unlikely	Possible	Likely	Very Likely
Enzmology	Individual Z	Z < 2	Z ≥ 2	Z ≥ 3	Z ≥ 4
	Sum + Z only	Z < 7	Z ≥ 7	Z ≥ 9	Z ≥ 11
Oxygraphy	Individual Z	Z < 3	Z ≥ 3	Z ≥ 4	Z ≥ 5
	Sum + Z only	Z < 10	Z ≥ 10	Z ≥ 15	Z ≥ 20

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
