# Peer review of "Oxygraphy Versus Enzymology for the Biochemical Diagnosis of Primary Mitochondrial Disease"

_metabolites, 2019, doi:10.3390/metabo9100220_

Round 1
Reviewer 1 Report
While reading your paper, I had so many questions that I thought would be answered in the Discussion, only to find the latter to be one of the shortest discussions I've ever seen in a major paper as you're trying to present it (you did discuss a few items in Results). I am not certain why I’d do oxygraphy as a clinician trying to diagnose PMD if you already had genetically confirmed patients. The whole idea is to avoid invasive studies so what would be more valuable to demonstrate its usefulness is to do oxygraphy before genetic studies and then see how they compare, preferrably in a blind fashion. The major question that you didn’t answer is how would oxygraphy help if I don’t have genetic confirmation? If you get “very likely” but no genetic diagnosis, are you going to treat it as PMD? Conversely, if you get genetic diagnosis but “unlikely” on oxygraphy, would you still treat the patient as PMD? If you described a blood test, I’d possibly see its use but an invasive skin biopsy just to get another piece of information that can be false positive in absence of genetic diagnosis (many “positive” muscle studies turn out to be non-PMD), is not entirely convincing. You also didn’t address if you’re going to do Whole Exome or even Whole Genome when oxygraphy and genetic data don’t agree. Or are you going to jump into PMD treatment based on oxygraphy and everything else but genetic diagnosis while realizing that there may be other process going on as oxygraphy is not specific to PMD. As a matter of fact, you state that enzymology has a higher specificity and yet, reports of specificity of even enzymology are “sparse.” Are you offering yet another one of a multitude of tests for PMD that adds very little to specificity? Please address these questions in the Discussion. In the field of PMD, we don’t yet have proven treatments, but we have too many tests that are costly, invasive and often misleading. Tell me why we need yet another invasive test.
Author Response
We thank reviewer one for your feedback. Broadly speaking, it would appear that we gave a false impression that we offered oxygraphy as a one stop test PMD, which was not our intention. Our aim was merely to examine the potential clinical utility of the oxygraphy method in diagnosing PMD, as compared to the standard enzymological method. Particularly in the discussion, we have tried to address these comments as described below.While reading your paper, I had so many questions that I thought would be answered in the Discussion, only to find the latter to be one of the shortest discussions I've ever seen in a major paper as you're trying to present it (you did discuss a few items in Results).
I am not certain why I’d do oxygraphy as a clinician trying to diagnose PMD if you already had genetically confirmed patients.
This is a retrospective study to assess the capabilities of the two techniques. To this end, genetically diagnosed patients were enrolled in the study such that we could accurately state the sensitivity of the technique in our cohort by our scoring method. We were not intending to suggest that in the case of a definitive genetic diagnosis that biochemical testing is required. To this end, we have significantly expanded the discussion to clarify this point, including through the inclusion of a diagnostic guide which clearly indicates that biochemical tools are most valuable where genetic advice is inconclusive.The whole idea is to avoid invasive studies so what would be more valuable to demonstrate its usefulness is to do oxygraphy before genetic studies and then see how they compare, preferrably in a blind fashion.
We have clarified predominantly in the discussion that this is a retrospective study in a genetically defined cohort of PMD patients to enable an assessment of the capabilities of the two techniques. Such an analysis is a necessary precursor to such a prospective study, which we are currently preparing to commence in suspected PMD patients in which targeted genetic analysis returned inconclusive results.The major question that you didn’t answer is how would oxygraphy help if I don’t have genetic confirmation? If you get “very likely” but no genetic diagnosis, are you going to treat it as PMD? Conversely, if you get genetic diagnosis but “unlikely” on oxygraphy, would you still treat the patient as PMD?
As above, we have modified the discussion to clarify that a definitive genetic diagnosis is the optimal guide as to the presence of a PMD. Further, neither oxygraphy or enzymological testing is capable of making a definitive diagnosis of a PMD, please read the inclusion at line 343: “It is also critical to note that our results, as defined though our diagnostic scoring matrix, are not intended to be applied as a definitive guide as to the presence of a PMD, but must be considered together with the clinical presentation of the patient, and any genetic results (see Figure 5 for suggested interpretation of enzymology and oxygraphy results in the context of variants of uncertain genetic significance).”.Our scoring system was designed as a guide to interpreting the results as implied by the phrase in the materials and methods “Dysfunction indicating disease”, and that our scale is from “unlikely” to “very likely”. At no point do we refer to “confirmed”.
If you described a blood test, I’d possibly see its use but an invasive skin biopsy just to get another piece of information that can be false positive in absence of genetic diagnosis (many “positive” muscle studies turn out to be non-PMD), is not entirely convincing.
We have investigated the suitability of blood for this purpose, but observed poor separation of patients from controls (data not discussed or referred to in the manuscript whose focus was entirely on fibroblasts). Indeed, we acknowledge that any biochemical test, as applied in any sample, for the diagnosis of PMD is prone to false negatives. To this end, please see the modified discussion as referred to above which addresses this point. Further, while a skin biopsy is technically invasive, it is hardly more invasive than taking a blood sample, and significantly less invasive than a liver or muscle biopsy that is commonly collected for enzymological testing. Similarly, the critical organ of the CNS cannot be directly biopsied in sufficient amounts for such biochemical testing, necessitating the analysis of other tissues or cell types. Again, please see the modified discussion.
You also didn’t address if you’re going to do Whole Exome or even Whole Genome when oxygraphy and genetic data don’t agree.
As above, biochemical testing is most valuable where a genetic diagnosis was not definitive (targeted analysis, whole exome or whole genome). Further, our manuscript is not intended to act as a definitive guide as the diagnostic process for PMD, but merely comment on the capabilities of the biochemical techniques of oxygraphy and enzymology in the situation where biochemical guidance is sought.Or are you going to jump into PMD treatment based on oxygraphy and everything else but genetic diagnosis while realizing that there may be other process going on as oxygraphy is not specific to PMD.
Again at line 343: “It is also critical to note that our results, as defined though our diagnostic scoring matrix, are not intended to be applied as a definitive guide as to the presence of a PMD, but must be considered together with the clinical presentation of the patient, and any genetic results (see Figure 5 for suggested interpretation of enzymology and oxygraphy results in the context of variants of uncertain genetic significance)”.As a matter of fact, you state that enzymology has a higher specificity and yet, reports of specificity of even enzymology are “sparse.”
We assume that the comment referred to line 348: “Here we find that while oxygraphy outperforms enzymology for both sensitivity and predictive strength (“unlikely” to “very likely”), the enzymological method provides more in-depth biochemical information about the nature of the defect, and has a slight advantage in specificity in our dataset”. Specificity here was used to refer to the nature of the underlying biochemical defect. However, we agree this was both confusing and redundant in the context of the preceding text. We have therefore amended this text to read: “Here we find that while oxygraphy outperforms enzymology for both sensitivity and predictive strength (“unlikely” to “very likely”), the enzymological method provides more in-depth biochemical information about the nature of the defect (eg., isolated or combined OXPHOS dysfunction; Figure 5)”
We have also corrected line 104 that previously read: “there are only scarce reports of their specificity”. This was an error, this should have referred to “sensitivity”, given this was the focus of our research in the context of a retrospective investigation.
Similarly, “Specificity” was incorrectly referred to in the materials and methods under the title “Diagnostic prediction”. We did not measure specificity, and this description has been removed.
Are you offering yet another one of a multitude of tests for PMD that adds very little to specificity? Please address these questions in the Discussion.
In regards to specificity, we were unable to evaluate this measure owing the retrospective nature of the study as previously described at line X: “Specificity is not delineated considering that the comparison group was healthy controls and not patients suspected of a mitochondrial disease with subsequent genetic and or functional indication of an unrelated disease.”
We do concur that there is not a lot of daylight between the sensitivity of the two techniques. However, this simple number hides that oxygraphy provides a much stronger disease prediction, is quicker, and can capture defects that enzymology classically does not (such as SLC25A42 deficiency) or requires additional enzymological testing beyond the OXPHOS system (such as PDHC deficiency). We have significantly elaborated on this in the discussion.
In the field of PMD, we don’t yet have proven treatments, but we have too many tests that are costly, invasive and often misleading. Tell me why we need yet another invasive test.
We need another test because fibroblasts are minimally invasive, and you detect 79% of patients in our cohort by the oxygraphy method, and 73% of these diagnoses made come with the label “very likely” by our method. That no proven treatments exist is largely an irrelevant point in a group of patients which require an end to a diagnostic odyssey. If genetics can’t end this odyssey, quality biochemical tools are required, and we proffer that oxygraphy appears to be one such quality tool. Why would you do a muscle or liver biopsy first when you could start with fibroblasts and have a strong indication (or not) of a PMD? Again, we have expanded on this point throughout the discussion.Reviewer 2 Report
In their manuscript, Bird et al. conduct respiratory studies in fibroblasts of patients with primary mitochondrial deficiency, and compare their outcome with biochemical assays, which were performed on these fibroblasts earlier. They conclude that in some scenarios respiratory studies demonstrate somewhat higher sensitivity, and when used in conjunction with biochemical studies may provide higher overall sensitivity as compared to either method alone. The experiments were performed at a good methodological level and are of interest to mitochondrial community. However, some substantial weaknesses were noted, specifically:
It has been acknowledged back in '90s that fibroblasts from PMD patients are only about 50% concordant with respect to respiratory defect (measured enzymatically) in primary tissue/organ affected. This has been attributed to tissue/organ specificity of PMD. The authors' own data confirm those early observations. To eliminate bias, these early observations must be acknowledged in the Introduction and addressed in the Discussion. Also, it appears imperative to explain the readership why the authors believe that fibroblasts are a good proxy for all other tissues (e.g., nervous, muscle) that are often affected by PMD. If PMD affects brain, why should we expect to always see a respiratory deficiency in cultured fibroblasts? How surprising is it not to see a defect in fibroblasts? Examples of disease/symptoms due to respiratory deficiency in fibroblasts in vivo would be very helpful.
Another aspect of studies in patient fibroblasts is that fibroblasts for these studies are taken from their natural environment and grown in 2D culture as opposed to 3D growth in vivo. During introduction into culture, they are adapted to growth under much higher partial oxygen tensions as compared to those encountered in vivo and in the presence of the altered nomenclature and concentrations of nutrients. Is it reasonable to expect that these alterations will have the same effect on genetically different fibroblast cultures (e.g., wt vs. patient)? Would you predict that these alterations would not affect respiration in these fibroblasts? If so, why? If not, what are the implications for assays performed in this study? These relevant issues have to be discussed.
Minor issues: certain statements in the paper make no sense and should be corrected or explained. E.g., the authors state that they examined fibroblasts from 34 patients, of which 20 were females and 13 males (20+13=33, not 34). Lanes 93 and 111, respectively. On the line 106 the authors state that "23 of these mutations were encoded for by nDNA, and 11 by mtDNA (32% and 68% respectively of our cohort). Actually, these percentages are counter to respective. On lines 181 and 182 48%+21%=69%, not 68%. Please, check. If needed, add decimals so that the math makes sense. KOlmogorov is spelled with an "O", not "A". Google is unaware what does "hides though" (line 181) mean (perhaps, rephrase?)
Author Response
It has been acknowledged back in '90s that fibroblasts from PMD patients are only about 50% concordant with respect to respiratory defect (measured enzymatically) in primary tissue/organ affected. This has been attributed to tissue/organ specificity of PMD. The authors' own data confirm those early observations. To eliminate bias, these early observations must be acknowledged in the Introduction and addressed in the Discussion.
Our data was limited to the analysis of fibroblasts, so we therefore were unable to make a direct comparison to diagnostic rates made by enzymology in primary tissues such as heart and muscle. We do therefore not agree that our results confirmed such earlier observations owing to our inability to make such a direct comparison. Further, despite an extensive literature review search, we were unable to identify the research mentioned. The existing reference to Rodenburg et al 2013 was the most comprehensive (and multi center) analysis that we could find, which compares diagnostic yield in muscle and fibroblasts in a limited number of patients (3 PMD patients for muscle and 16 fibroblasts, but not paired). Going back to the 90’s, many isolated reports of enzyme deficiencies exist, but broad screening was scarce. Work led by A Munnich, (PMID 7955428 – enzymology performed in 10 muscle samples from PMD patients) and P. Lestienne (PMID: 2156958 – 9 PMD myopathy patients muscle biopsies were analysed) was the most extensive that we could find. None of these papers however gave a clear indication of the diagnostic yield of primary tissues compared to fibroblasts, nor the sensitivity of the technique as applied in fibroblasts (Rodenburg et al 2013 came the closest with its panel of 16 fibroblast cell lines). Additional work in control samples again directed by A Munnich was extensive, however only referred to data not shown for patients, or referenced smaller studies when describing the clinical utility of the enzymological method in muscle and fibroblasts (PMID: 7955429). If the reviewer would be kind enough as to identify the literature in question, we would be glad to reference and acknowledge relative to our research.
Please see our modified introduction at line 34: “Surprisingly, although these methods have been well described for many years [8-11], the sensitivity and specificity of enzymological testing for the diagnosis of PMD is only sparsely reported in the literature as assessed in a panel of patients with defects of diverse genetic origins and pathophysiology. From the limited data available, the sensitivity in fibroblasts was in the order of 75% [12] and higher in muscle at 80-100% [12-14], albeit these numbers are drawn from limited sample sizes and are not paired in their analysis”
Also, it appears imperative to explain the readership why the authors believe that fibroblasts are a good proxy for all other tissues (e.g., nervous, muscle) that are often affected by PMD. If PMD affects brain, why should we expect to always see a respiratory deficiency in cultured fibroblasts? How surprising is it not to see a defect in fibroblasts? Examples of disease/symptoms due to respiratory deficiency in fibroblasts in vivo would be very helpful.
Fibroblasts were not chosen for their relevance to disease, but rather for their availability. They are then surely not the ideal sample type to test for PMD, nor clinically relevant cell type in disease, but rather a readily available cell type in which mitochondrial function can be monitored and compared to control fibroblasts for diagnostic purposes.See new entry in the discussion at line 329: “Fibroblasts were chosen as they are available from a minimally invasive biopsy and can be readily maintained in culture, as compared to the preferred liver and muscle for enzymology which are invasively obtained. It should be further considered that tissue from the commonly affected organ in PMD, the CNS, cannot be directly tested at all. While we acknowledge that skin disease is not a common presentation in PMD (most typically heart, muscle, liver, kidney or CNS dysfunction), fibroblasts are utilised here in a diagnostic context to indicate the presence of a PMD or not, and therefore do not provide insights into the organ specific nature of the disease … Considering the diverse clinical presentations in PMD, it is also unlikely that any one tissue or sample type will reliably display impaired mitochondrial function, thus complicating the biochemical diagnosis of these diseases.”
Another aspect of studies in patient fibroblasts is that fibroblasts for these studies are taken from their natural environment and grown in 2D culture as opposed to 3D growth in vivo. During introduction into culture, they are adapted to growth under much higher partial oxygen tensions as compared to those encountered in vivo and in the presence of the altered nomenclature and concentrations of nutrients. Is it reasonable to expect that these alterations will have the same effect on genetically different fibroblast cultures (e.g., wt vs. patient)? Would you predict that these alterations would not affect respiration in these fibroblasts? If so, why? If not, what are the implications for assays performed in this study? These relevant issues have to be discussed.
While fibroblasts in culture may not mimic the in vivo environment, we consider this irrelevant in the context of a diagnostic. Simply put - can we see a difference to controls or can we not? Please see adaption to this effect in the discussion line 335: “Similarly, we acknowledge that our PMD fibroblasts are maintained in a 2-D culture environment, in which oxygen saturation and mitochondrial function are altered from their physiological state. However, these variables are equally relevant for control and PMD fibroblasts, and as viewed through a diagnostic prism have little relevance as we merely seek to distinguish PMD fibroblasts from non-PMD fibroblasts.”Minor issues: certain statements in the paper make no sense and should be corrected or explained. E.g., the authors state that they examined fibroblasts from 34 patients, of which 20 were females and 13 males (20+13=33, not 34).
Thank you – yes, 14 males. CorrectedLanes 93 and 111, respectively. On the line 106 the authors state that "23 of these mutations were encoded for by nDNA, and 11 by mtDNA (32% and 68% respectively of our cohort). Actually, these percentages are counter to respective.
Yes, they were the wrong way around. CorrectedOn lines 181 and 182 48%+21%=69%, not 68%. Please, check. If needed, add decimals so that the math makes sense.
This was an error, the “likely” and “very likely” groups comprised 47% of this cohort. CorrectedKOlmogorov is spelled with an "O", not "A".
Google is unaware what does "hides though" (line 181) mean (perhaps, rephrase?)
Please see modified text at line 152: “Contained within this number though is that only 47% of these patients were either characterised as either “likely” or “very likely”, with a further 21% only as “possible”.”Reviewer 3 Report
In this study by Bird et al., the authors apply the technique of oxygraphy to primary mitochondrial disease fibroblast cells, and compare the techniques to enzymology techniques that are the standard for diagnosing PMD. This manuscript is well-written and clearly presented. The data generally support the conclusions made. The authors are to be commended for conceptualizing and performing this study, which, in the estimation of this reviewer, will be an important contribution to both the basic and clinical science PMD fields. As described by the authors, the combinatorial approach of enzymology and oxygraphy appears to provide a more nuanced and well-rounded image of the biological deficits in PMD patient cells. The authors provide a good overview of the limitations of current enzymlogical-based analyses of PMD patient cells, and accurately describe the advantages (and disadvantages) of adding oxygraphy to the current standard.
I recommend that this manuscript can be accepted without any required edits. However, I present the following minor suggestions that might enhance the impact or clarity of the study. It is up to the authors' discretion whether or not they wish to address or act on any of these suggestions:
1) The authors could more clearly describe why they chose their particular scoring system compared to other scoring systems.
2) While the authors are to be commended for plainly describing the limitations of oxygraphy, some of these descriptions come across as overly negative, when they do not need to be negative. For example, lines 222-234 could alternatively be described as the authors overcoming the variability inherent in oxygraphy measurements by performing technical triplicates for each sample.
3) The discussion section could be expanded to include a more detailed analysis of the potential strenghts and weaknesses of each individual assay and the combinatorial approach. For example, the authors could expand on the important conclusion that they allude to in lines 184-188 which demonstrate that the oxygraphy approach may identify more biologically relevant OXPHOS issues than enzymology can due to the intact state of the cells assessed.
Author Response
1) The authors could more clearly describe why they chose their particular scoring system compared to other scoring systems.
Please see included text in the materials and methods at line 453: “Z score thresholds were chosen for optimal separation of controls and PMD fibroblasts, with a step wise increase in Z scores indicating to us an increased likelihood of mitochondrial dysfunction and thus likeliness of a PMD.”2) While the authors are to be commended for plainly describing the limitations of oxygraphy, some of these descriptions come across as overly negative, when they do not need to be negative. For example, lines 222-234 could alternatively be described as the authors overcoming the variability inherent in oxygraphy measurements by performing technical triplicates for each sample.
We were wary of over selling the technique and not fully recognizing its limitations in comparison to the enzymological method. However, we are happy to make this change.Please see in results at line 208: “Reducing the technical variation was surmounted in our hands by performing 3 technical replicates per sample.”
3) The discussion section could be expanded to include a more detailed analysis of the potential strenghts and weaknesses of each individual assay and the combinatorial approach. For example, the authors could expand on the important conclusion that they allude to in lines 184-188 which demonstrate that the oxygraphy approach may identify more biologically relevant OXPHOS issues than enzymology can due to the intact state of the cells assessed.
We agree that this is a major advantage of this method, and have included this sentiment in the discussion as suggestedPlease read at line 318: “The oxygraphy technique was chosen as unlike enzymology, the assay is relatively quick to perform, and provides broader insight into mitochondrial function than just the OXPHOS system.”
And at line 351: “In favour of the oxygraphy method, however, is implicit in the intact nature of the cells in the assay, and the reliance on multiple mitochondrial functions for respiratory activity. In this regard, oxygraphy has the capacity to detect defects in and closely associated with the TCA cycle, which are otherwise not captured by enzymology.”
Round 2
Reviewer 1 Report
The authors did not adequately address the difference between PMD and secondary mito dysfunction (SMD). Even though you aim to describe oxymetry in PMD, you do mention that some effects are secondary. How does oxygraphy help distinguish between PMD and SMD which is important for diagnostic and management reasons - please see PMID: 27587988. You should discuss this in discussion without necessarily trying to come up with the new data. You should show the importance of PMD vs SMD or don't mention secondary at all which will be a limitation of the study.
Author Response
The authors did not adequately address the difference between PMD and secondary mito dysfunction (SMD). Even though you aim to describe oxymetry in PMD, you do mention that some effects are secondary. How does oxygraphy help distinguish between PMD and SMD which is important for diagnostic and management reasons - please see PMID: 27587988. You should discuss this in discussion without necessarily trying to come up with the new data. You should show the importance of PMD vs SMD or don't mention secondary at all which will be a limitation of the study.
We agree that distinguishing PMD from SMD is an ongoing challenge. Having not assessed this in our own study, we are unfortunately unable to examine the discriminatory power of oxygraphy in this regard. We have however now recognised this group of patients, and acknowledged the limitations of our own study in discerning between PMD and SMD patients by the enzymology and oxygraphy methods that we employed.
Please see in the introduction at line 20: “Further complicating their diagnosis are patients with secondary mitochondrial disease (SMD) [8]. SMD patients appear phenotypically and biochemically similar to PMD patients, but their underlying basis is non-mitochondrial. SMD patients have their roots in mutations in non-mitochondrial genes (eg. ATP7B mutations causing Wilson’s disease) or environmental factors. While distinguishing PMD from SMD patients then is an ongoing challenge, this was not assessed in our own study.”
And in the discussion at line 308: “The complex clinical presentations and molecular basis of PMD, and their phenotypic and biochemical overlap with SMD, is an ongoing challenge in their diagnosis, and many patients experience a long and conflicting diagnostic odyssey [4]”
AND at line 354: “By translation however, while not tested in this analysis, oxygraphy may also be more likely to detect mitochondrial dysfunction in SMD patients than by direct testing of the OXPHOS system using enzymology, thus causing the specificity of diagnosis made by the oxygraphy method to be diminished relative to enzymology.”
Reviewer 2 Report
I appreciate the difficulty of locating specific references based on a loose description as I experienced that myself. Nevertheless, the issue of tissue-specific manifestation of respiratory chain disorders is not a fantasy. A recent study noted that "The use of the correct tissue for assessing complex I activity is also essential. Many complex I deficiencies expressed in the muscle are not expressed in fibroblasts.44 Of 87 complex I patients in this study for whom multiple tissues were studied, the enzyme defect was tissue specific in 36 (41%). In most of these patients, the defect was not expressed in fibroblasts, but in 13 patients, complex I was normal in the skeletal muscle but deficient in the liver or heart. " PMID: 21364701. I believe that this issue should be addressed in the manuscript.
While it is agreeable that nonphysiological cell culture conditions are the same for both control and patients' fibroblasts, it would be incorrect to assume that the effects of these conditions would be the same for both types of cells. Hypothetically speaking, if mitochondrial defect stems from a reduced affinity to oxygen, then culturing/assaying cells at ambient oxygen concentration may selectively rescue this defect in patient fibroblasts without having effect (or having a minimal one) on wt fibroblasts because of their high affinity/low Km for oxygen.
Author Response
I appreciate the difficulty of locating specific references based on a loose description as I experienced that myself. Nevertheless, the issue of tissue-specific manifestation of respiratory chain disorders is not a fantasy. A recent study noted that "The use of the correct tissue for assessing complex I activity is also essential. Many complex I deficiencies expressed in the muscle are not expressed in fibroblasts.44 Of 87 complex I patients in this study for whom multiple tissues were studied, the enzyme defect was tissue specific in 36 (41%). In most of these patients, the defect was not expressed in fibroblasts, but in 13 patients, complex I was normal in the skeletal muscle but deficient in the liver or heart. " PMID: 21364701. I believe that this issue should be addressed in the manuscript.
We appreciate you identifying the research alluded to. Unfortunately, however, while the analysis in the passage provided may be correct, the reference contained in this quote refers to PMID: 8884581, “Genetic counselling and prenatal diagnosis in disorders of the mitochondrial energy metabolism”. This article does not contain any data which resembles that referred to in PMID: 21364701 as quoted above. We also searched the surrounding references in this article, but none identified the data under discussion. We further searched pubmed and google scholar using the limited terms which may have allowed us to identify the data, but were unsuccessful. Finally, we contacted the last author of PMID: 21364701 to identify the correct citation, but had not received a response at the time of submitting this response. As previously discussed in our previous response, these topics are also discussed in PMID: 7955429, such as “The usefulness of lymphocyte investigation as a first step in the investigation of RC disorders, is illustrated by the results from our study. In 42 patients with a RC deficiency in whom both lymphocytes and skeletal muscle activities were investigated, we found that 50% expressed the deficiency in both tissues, 45% had only a muscular expression and 5% expressed the deficiency only in lymphocytes.”, however, no supporting data is displayed or referenced. As before, we would be very pleased to refer to this data if it can be identified so it can be cited specifically. Despite PMID: 21364701 (reference 16 below) being a poor reference then for this topic, we have acknowledged this point in the introduction at Line 44: “Nonetheless, it is critical then to consider that fibroblasts as employed in our own study may not be the optimal sample for the diagnosis of PMD, and that PMD manifests with organ specific pathophysiology [16]. This is not to discount that mitochondrial function is likely impaired in these other unaffected organs and tissues, but merely that there is no disease manifestation.”While it is agreeable that nonphysiological cell culture conditions are the same for both control and patients' fibroblasts, it would be incorrect to assume that the effects of these conditions would be the same for both types of cells. Hypothetically speaking, if mitochondrial defect stems from a reduced affinity to oxygen, then culturing/assaying cells at ambient oxygen concentration may selectively rescue this defect in patient fibroblasts without having effect (or having a minimal one) on wt fibroblasts because of their high affinity/low Km for oxygen.
We agree that such effects may influence the sensitivity of our assays and have amended the discussion accordingly. Please see at line 331: “Similarly, we acknowledge that our PMD fibroblasts are maintained in a 2-D culture environment, in which oxygen saturation and mitochondrial function are altered from their physiological state. While these variables are equally applicable for control and PMD fibroblasts, we cannot discount effects that may disproportionally favour mitochondrial function in PMD cells and so reduce the diagnostic sensitivity of both the enzymology and oxygraphy methods. For example, oxygen partial pressure is significantly higher in an in vitro cell culture setting (19%) compared to 2-5% in in vivo tissues [20]. Accordingly, in the situation that PMD arises due to reduced affinity for oxygen, higher oxygen partial pressures in the in vitro culture system may rescue (or partially) the defect without influencing control cells function, thus masking underlying dysfunction in cultured PMD fibroblasts.”Round 3
Reviewer 1 Report
thank you for describing the limitations when it comes to SMD
Reviewer 2 Report
I accept the authors' response.